Exploring the binding properties and structural stability of an opsin in the chytrid Spizellomyces punctatus using comparative and molecular modeling

Ahrendt Steven R. sahrendt0@gmail.com sahrendt0@berkeley.edu 1 2 3
Medina Edgar Mauricio 4 5
Chang Chia-en A. 2 6
Stajich Jason E. jason.stajich@ucr.edu 1 2
1 Department of Plant Pathology & Microbiology, University of California , Riverside , CA , USA
2 Institute for Integrative Genome Biology, University of California , Riverside , CA , USA
3 Genetics, Genomics, and Bioinformatics Program, University of California , Riverside , CA , USA
4 Department of Biology, Duke University , Durham , NC , USA
5 Center for Genomic and Computational Biology, Duke University , Durham , NC , USA
6 Department of Chemistry, University of California , Riverside , CA , USA
Silva Pedro
Electronic publication date: 2017 Apr 27
Publication date: 2017
Volume: 5
Electronic Location ID: e3206
Received 2016 Sep 1; Accepted 2017 Mar 20
Copyright: ©2017 Ahrendt et al.
Copyright year: 2017
Copyright holder: Ahrendt et al.
License: This is an open access article distributed under the terms of the Creative Commons Attribution License, which permits unrestricted use, distribution, reproduction and adaptation in any medium and for any purpose provided that it is properly attributed. For attribution, the original author(s), title, publication source (PeerJ) and either DOI or URL of the article must be cited.
License URL: https://creativecommons.org/licenses/by/4.0/

Keywords: Chytrid, Opsin, Homology modeling, Light receptor, Protein structure, GPCR, Early diverging fungi, Evolution, Mycology, Zoosporic

Funding: Alfred P. Sloan Foundation University of California-Riverside College of Agriculture and Natural Sciences USDA National Institute of Food and Agriculture Hatch CA-R-PPA-5062-H NSF MRI DBI-1429826 NIH S10-OD016290 Jason E. Stajich and Steven R. Ahrendt were supported by funds from a grant from Alfred P. Sloan Foundation to J.E.S., initial complement funds to J.E.S from the University of California-Riverside College of Agriculture and Natural Sciences, and USDA National Institute of Food and Agriculture Hatch project CA-R-PPA-5062-H. High performance computing resources on the UC Riverside Institute for Integrative Genome Biology cluster were supported by NSF MRI DBI-1429826 and NIH S10-OD016290 grants. The funders had no role in study design, data collection and analysis, decision to publish, or preparation of the manuscript.

==============================
Background

Opsin proteins are seven transmembrane receptor proteins which detect light. Opsins can be classified into two types and share little sequence identity: type 1, typically found in bacteria, and type 2, primarily characterized in metazoa. The type 2 opsins (Rhodopsins) are a subfamily of G-protein coupled receptors (GPCRs), a large and diverse class of seven transmembrane proteins and are generally restricted to metazoan lineages. Fungi use light receptors including opsins to sense the environment and transduce signals for developmental or metabolic changes. Opsins characterized in the Dikarya (Ascomycetes and Basidiomycetes) are of the type 1 bacteriorhodopsin family but the early diverging fungal lineages have not been as well surveyed. We identified by sequence similarity a rhodopsin-like GPCR in genomes of early diverging chytrids and examined the structural characteristics of this protein to assess its likelihood to be homologous to animal rhodopsins and bind similar chromophores.

Methods

We used template-based structure modeling, automated ligand docking, and molecular modeling to assess the structural and binding properties of an identified opsin-like protein found in Spizellomyces punctatus, a unicellular, flagellated species belonging to Chytridiomycota, one of the earliest diverging fungal lineages. We tested if the sequence and inferred structure were consistent with a solved crystal structure of a type 2 rhodopsin from the squid Todarodes pacificus.

Results

Our results indicate that the Spizellomyces opsin has structural characteristics consistent with functional animal type 2 rhodopsins and is capable of maintaining a stable structure when associated with the retinaldehyde chromophore, specifically the 9-cis-retinal isomer. Together, these results support further the homology of Spizellomyces opsins to animal type 2 rhodopsins.

Discussion

This represents the first test of structure/function relationship of a type 2 rhodopsin identified in early branching fungal lineages, and provides a foundation for future work exploring pathways and components of photoreception in early fungi.

Introduction

An organism experiences a multitude of environmental stimuli including chemical, gravity, the Earth’s magnetic field, pressure, and light. The biochemical ability to appropriately process and respond to these signals is a complex and involved task, and understanding the molecular mechanisms of these responses is an ongoing scientific challenge. The presence or absence of light is perhaps one of the easiest sources of stimuli to comprehend and observe. The daily cycles of sunlight due to the rotation of the planet has had such a profound influence on the development of life that it comes as no surprise to find some form of photoreception in nearly every organism on the planet. The widespread occurrence of such an ability, however varied in its implementation, speaks to its importance during the earliest stages of development of life.

In Fungi, there are several classes of proteins capable of photoreception that function by different mechanisms of action and have varied structures, sensitivities, and specializations. These include blue light responsive white-collar complex, VIVID and cryptochrome photoreceptors, red light responsive phytochromes, and multi-wavelength light responsive opsins (Idnurm, Verma & Corrochano, 2010). The opsins are a large class of seven-transmembrane proteins which bind retinylidene compounds required for photoreception and can be subdivided into Types 1 or 2 based on phylogenetic history, sequence similarity, and function. The classes share some characteristics in structure (i.e., seven helical transmembrane domains) and mechanism of activation (i.e., photoisomerization of a retinaldehyde chromophore) but have distinct evolutionary histories (Spudich et al., 2000).

Opsins are part of the large G-protein coupled receptor (GPCR) superfamily, which has more than 800 distinct described members in humans (Lagerström & Schiöth, 2008). GPCR proteins share a similar architecture: seven membrane-spanning helical regions connected by three intracellular and three extracellular loop regions. The cytoplasmic region of the GPCR interacts with heterotrimeric G proteins found on the intracellular side of the plasma membrane, which in turn function in signal transduction (Neves, Ram & Iyengar, 2002). Of the five major GPCR families, the Class-A family, comprising the opsins, various neurotransmitters, and hormone receptors, is by far the largest with approximately 700 proteins classified into four subfamilies (Katritch, Cherezov & Stevens, 2013).

The “Type 2 rhodopsins” represent a small subgroup of the opsin subfamily of Class-A GPCRs. Unlike other members of this class, they are activated by the interaction between a single photon of light and a covalently bound chromophore. A functional rhodopsin (rhodopsin pigment) is generated when an opsin apoprotein forms a covalent bond with a retinaldehyde chromophore via a Schiff-base linkage at a conserved lysine residue. While 11-cis-retinal is the most common chromophore observed in vertebrates and invertebrates, additional types are also found in nature. For example, 3,4-dehydroretinal is observed in fish, amphibians, and reptiles. Switching between the 11-cis- and 3,4-dehydro-chromophores can be employed as a light adaptation strategy in certain freshwater fish (Shichida & Matsuyama, 2009). 3-hydroxyretinal is found in insects, while 4-hydroxyretinal is observed in the firefly squid. In addition to the 11-cis-conformation, retinal can adopt a number of different isomers, including all-trans-, 13-cis-, and 9-cis- (Shichida & Matsuyama, 2009). Previous studies using hybrid quantum mechanics/molecular mechanics (QM/MM) simulations suggest that the 11-cis-retinal isomer has been evolutionarily selected as the optimal chromophore due to the energetic stability of the resulting chromophore-opsin pigment (Sekharan & Morokuma, 2011).

Activation of the rhodopsin occurs through the photoisomerization of 11-cis-retinal to all-trans-retinal, which causes a conformational change in the protein structure of the receptor. Alternatively, the ion transporter rhodopsins (part of the “Type 1 opsins”) are activated by the photoisomerization of all-trans-retinal to 13-cis-retinal. These function as membrane channels and are typically used for light-driven membrane depolarization via proton or chloride ion pumping. Examples of this group can be found in bacteria, archaea, and eukaryotes, and include the bacterial sensory rhodopsins, channelrhodopsins, bacteriorhodopsins, halorhodopsins, and proteorhodopsins (Zhang et al., 2011). The nature of the evolutionary relationship between the two types of rhodopsin has not been definitively established and is currently the subject of discussion (Terakita, 2005; Shichida & Matsuyama, 2009; Becker et al., 2016; Devine, Theobald & Oprian, 2016).

There are at least three distinct phyla of early diverging fungi which are often referred to as “zoosporic fungi” or, more informally, “chytrids”: the Cryptomycota, Chytridiomycota, and Blastocladiomycota (James et al., 2006; Stajich et al., 2009; Jones et al., 2011; James et al., 2013). These lineages share as a defining characteristic the presence of a flagellated life stage called a zoospore. Previous work has demonstrated that some species in these early diverging lineages are phototaxic. For example, the marine chytridiomycete Rhizophydium littoreum will respond to light at a variety of wavelengths, with the most rapid response occurring at 400 nm (Muehlstein, Amon & Leffler, 1987). While the evidence strongly suggests blue-light sensitivity, the researchers did not specifically characterize the active photoreceptor. Similarly, zoospores from the blastocladiomycete Allomyces reticulatus were determined not only to be phototactic, but also to possess visible, red-pigmented eyespots in which the photosensitive proteins are localized (Saranak & Foster, 1997). Careful analysis determined that the action spectrum of the phototactic A. reticulatus zoospores peaks at 536 ± 4 nm, similar to that of the human green-sensitive cone. More recently, comprehensive work on the related blastocladiomycete Blastocladiella emersonii demonstrated that a type 1 rhodopsin is in part responsible for phototaxis in response to green light (522 nm) (Avelar et al., 2014).

An initial analysis of the chytrid Batrachochytrium dendrobatidis genome revealed a surprising finding of a GPCR protein with similarity to the rhodopsin superfamily. Searches for additional genomes of early diverging fungi, including the saprotrophic chytrid Spizellomyces punctatus, revealed the presence of rhodopsin-like proteins in multiple zoosporic fungal lineages. The availability of these examples of opsin homologs in the deeply diverging fungal lineages suggested the shared ancestry of these light sensing receptors and the presence of this pathway in the fungal-animal ancestor (Krishnan et al., 2012; EM Medina, 2016, unpublished data).

The growing availability of X-ray structures of different GPCRs has illustrated a strong similarity in overall topology (Katritch, Cherezov & Stevens, 2013). As a result, structural models built for various GPCRs have been successful in in silico screening of inhibitors or examining protein dynamics (Bermudez & Wolber, 2015; Taddese et al., 2013; Ai & Chang, 2012). Comparative modeling, also known as “homology modeling”, is a computational method for building a structure for a protein of interest for which the structure is unknown. It is a template-based method which acts on the target’s sequence similarity to proteins for which the structure has been experimentally verified (template) (Sali, 1995). It is distinct from ab initio or de novo modeling, which instead uses only the target sequence and free-energy minimization techniques (Bradley, Misura & Baker, 2005). Homology modeling works best when there is high sequence identity between the target and template. Protein targets with sequence identity levels <30% with their template structure are often referred to as being in the “Twilight zone” of homology modeling, where models generated from these alignments are not of the highest quality (Chung & Subbiah, 1996). Coupled with molecular dynamics (another computational technique used to simulate interactions of complex molecules at the atomic level) and molecular docking (used to simulate protein-ligand interactions), homology modeling has multiple applications including structure-based drug discovery and investigations of protein dynamics.

The opsin-like proteins identified in the genomes of early diverging chytrid fungi are sufficiently similar to experimentally verified animal opsin structures for modeling and hypothesis testing about the potential ligand binding. We selected the Spizellomyces punctatus opsin-like GPCR for investigation as it possessed a conserved lysine residue suitable for retinal binding, unlike those in other chytrids. The target sequences and the rhodopsin homologs were modeled with Type 2 rhodopsin crystal structure templates made possible by the growing number of GPCR structures from the rhodopsin subfamily in the PDB (Katritch, Cherezov & Stevens, 2013). We generated a homology model for an opsin-like GPCR identified in the S. punctatus and use it to explore the binding properties of retinal isomers, the functional chromophores in rhodopsin-mediated photosensing. Here we show that the S. punctatus opsin is structurally similar to functional animal type 2 rhodopsins and is stable when associated with a 9-cis-retinal chromophore.

Materials & Methods

Sequence identification and homology modeling

Putative rhodopsin sequences in early diverging fungal lineages were identified based on sequence similarity to the Profile Hidden Markov model from the Pfam database (Finn et al., 2014), accession PF00001 (“7tm_1”). The HMM was searched against the predicted proteins from S. punctatus, B. dendrobatidis, and A. macrogynus HMMER v3.0 (Eddy, 2011) using e-value cutoff 1e−10. Inspection of the protein sequence of the S. punctatus homolog revealed a putative truncation, which lead us to correct the gene model at locus SPPG_00350 by adding a missing cytosine in the genome at position 1041 of the locus. The discrepancy was identified using exonerate (Slater & Birney, 2005) alignment of chytrid proteins to the genome to identify and correct this single deletion in the genome assembly (Supplemental Information 1; https://github.com/stajichlab/chytropsin). The amended protein sequence SPPG_00350T0L was used for subsequent analyses. The S. punctatus protein structure model was constructed using the I-TASSER server with the provided GPCR specific library (Zhang & Zhang, 2010). The normalized z-scores, indicative of alignment quality, of the top ten threading templates used by I-TASSER are provided in Table S2. Additionally, manual correction of the K320 orientation was performed by energy minimization using the general Amber force field (GAFF) (Wang et al., 2004) in Avogadro (Hanwell et al., 2012) after automatic refinement with OpusROTA (Lu, Dousis & Ma, 2008). The optimal model was selected using the I-TASSER provided “c-score”, a confidence value based on the significance of threading template alignments. The rhodopsin crystal structure from Todarodes pacificus (PDBid 2Z73; Murakami & Kouyama, 2008) was additionally selected for subsequent docking and molecular dynamics experiments. Stereochemical properties of both protein structures were validated using PROCHECK (v3.5) (Laskowski et al., 1993; Wiederstein & Sippl, 2007), ProQM (Ray, Lindahl & Wallner, 2010), and Verify3D (Lüthy, Bowie & Eisenberg, 1992). The S. punctatus homology model structure file is available on Github at http://github.com/stajichlab/chytropsin/.

Docking and Molecular dynamics (MD)

Automated protein-ligand docking was accomplished using Autodock 4 (Morris et al., 2009) and implementing a Lamarckian genetic algorithm approach for calculating the minimum free energy of binding of small molecules. Small molecule files were obtained from PubChem (Bolton et al., 2008) for the following isomers of retinal: 11-cis (A1), all-trans, 9-cis, 13-cis, 3,4-dehydro (A2), 3-hydroxy (A3), and 4-hydroxy (A4) used in the covalent docking screen. A covalent linkage was formed by manually specifying the presence of a bond between the terminal carbon atom in retinal and terminal nitrogen atom in the lysine side chain. The specific lysine predicted to be involved in Schiff-base linkage with the chromophore was inferred through multiple sequence alignment.

The dynamics of both the Todarodes and Spizellomyces rhodopsin complexes were investigated using all-atom molecular dynamics simulations with the Amber14 suite of programs (Case et al., 2015). Due to the computational expense of an explicit solvation model for simulating water molecules, an implicit solvation model (Onufriev, Bashford & David, 2000) (modified from the generalized Born solvation model Bashford & Case, 2000) was used in AMBER with the igb = 2 flag. The all atom force-field ff14SB (Hornak et al., 2006) was used as implemented in AMBER14, and GAFF was implemented for the ligand. Additionally, in order to mimic a membrane in which the protein would be found in vivo, all residues belonging to the transmembrane helices, except those within the binding pocket, were restricted using the restraint flag. Initial minimization was performed for 1 ns, followed by three NVT equilibration steps for 50 ps progressing from 200 K to 250 K to 298 K. The final production simulation was run for 100 ns at 298 K. For comparison, the photoisomerization of 11-cis-retinal to all-trans configuration occurs on the order of 200 fs (Smith, 2010).

For simulations of the squid structure, PDBid 2Z73 was used along with the structure of 11-cis-retinal crystallized with it. For the S. punctatus structure, simulations were performed using 9-cis-retinal ligand in the lowest energy conformation. 9-cis-retinal was chosen based on the covalent docking screen results. Backbone atoms were kept rigid while binding pocket residues were made flexible. Trajectory visualization and RMSD analysis were accomplished using VMD (v1.9.1) (Humphrey, Dalke & Schulten, 1996). Potential energy of the system was summarized using the process_mdout.perl script, provided with the AMBER package.

Results

Structural quality of homology model

For this study, a template-based model was constructed for the S. punctatus protein sequence using the I-TASSER website and GPCR specific database. Top Blastp hits of the S. punctatus protein to the PDB (as of 2016) include numerous opsin proteins, with the top scoring hit at 22.5% identity to a rhodopsin from Bos taurus (Table S1). Templates predicted by I-TASSER included both chains of the T. pacificus rhodopsin protein (Table S2). The S. punctatus protein shares 22% sequence identity with the T. pacificus sequence and several key functional and structural motifs are conserved between the structures (Fig. 1).

Figure 1 Structural details of the S. punctatus homology model.

(A) Structural alignment of S. punctatus homology model (grey) with T. pacificus crystal structure (light purple). S. punctatus residues are colored according to function: orange (binding pocket residues), red (putative counterion), purple (disulfide bond), yellow (salt bridge), dark blue (NPxxY motif), and pink & black (ion lock). Inset figures provide details for structural alignments of S. punctatus and T. pacificus (B) disulfide bond and salt bridge regions, (C) binding pocket residues, and (D) ERY and NPxxY regions.

The binding pocket comprises a number of hydrophobic residues which provide a sterically restrictive space in which the retinal ligand is situated (orange). The major functional residues in this group are the conserved lysine (cyan) and counterion (red) which facilitate proton transfer during photoisomerization. The ionic lock motif contains an (E/D)RY and NPxxY motif, which together act as a structural support which stabilizes the protein in the inactive (“dark”) state, and is broken upon receptor activation (Smith, 2010). In S. punctatus, the (E/D)RY and NPxxY motifs are both functionally conserved as 115ERY117 and 326NPVLF330 (pink). Two additional linkages are responsible for correct protein folding: a conserved disulfide bond between C110 and C187, and a conserved salt bridge between R117 and D190. S. punctatus model possesses both of these motifs as C91–C166 (yellow), and potentially R158–D169 (purple).

The quality of the S. punctatus homology model was assessed with Ramachandran plots (Ramachandran, Ramakrishnan & Sasisekharan, 1963), generated using PROCHECK (Laskowski et al., 1993; Wiederstein & Sippl, 2007), which graphically display the backbone dihedral angles (φ and ψ) of each amino acid residue in a protein. An aggregate assessment of observed protein structures determined by X-ray crystallography defines regions of acceptable stereochemistry; here using observed phi-psi distribution for 121,870 residues from 463 known X-ray protein structures. In practice, this analysis can be used for structure validation. A model with more than 90% of its residues having favorable stereochemistry is considered to be of good quality. For S. punctatus, the percentage of residues which fell within the most favorable region was 85.4%. The T. pacificus crystal structure of rhodopsin (Murakami & Kouyama, 2008) has a score of 90.9% in this category (Fig. S1).

Additionally, Verify3D (Lüthy, Bowie & Eisenberg, 1992) was used to assess model quality. Structures modeled correctly will have higher scores than structures which have been modeled incorrectly. Here, the S. punctatus model generated using the I-TASSER + GPCR database had a final score of 72.41, and 46.32% of the residues had an averaged 3D-1D score ≥0.2. For comparison, the rhodopsin X-ray crystal structure from T. pacificus had a final score of 87.85, and 58.86% of residues had a profile score ≥0.2. To provide further support that the S. punctatus model was constructed correctly, a model was generated with the S. punctatus sequence using the sensory rhodopsin II X-ray crystal structure from the archaeon Natronomonas pharaonis (PDBid 1H68, Royant et al., 2001), a type 1 opsin and thus a presumed incorrect modeling target. In this reconstruction, the final score was 15.08, and only 19.57% of residues had a Verify3D score ≥0.2. When the scores for these proteins are plotted as a function of their sequences (Fig. S2) the average scores fall between −0.12 and 0.66 (Fig. S2B) and −0.19 and 0.87 (Fig. S2A). The average scores for the S. punctatus structure model constructed against 1H68 however fall between −0.56 and 0.49 (Fig. S2C).

Finally, ProQM (Ray, Lindahl & Wallner, 2010) was used to assess model quality, providing a score between 0 (poor) and 1 (correctly modeled). The S. punctatus model had a global quality score of 0.5 and a range of local quality scores 0.03 to 0.91, with low scores corresponding to loop regions (Fig. S3A). The T. pacificus crystal structure had in general higher local quality scores, with a range of 0.11–1.13 and a global quality score of 0.766 (Fig. S3B). The quality assessment of the presumed mis-modeled S. punctatus homology model (described above) again suggested it was poorly modeled, with a global quality score of 0.42 and a range of local quality scores from −0.18 to 0.96 (Fig. S3C).

Computational ligand screen

Rhodopsin functions through the use of a retinaldehyde chromophore. The most common chromophore observed in both invertebrates and vertebrates is 11-cis-retinal (Shichida & Matsuyama, 2009). This retinal isomer is also used in the T. pacificus rhodopsin association. To determine if the S. punctatus rhodopsin utilized the same isomeric configuration of retinal, computational protein-ligand docking was performed using Autodock 4 with 11-cis-retinal and other vitamin-A based retinaldehyde compounds. The compounds 11-cis-retinal, all-trans-retinal, 9-cis-retinal, 13-cis-retinal, 3,4-dihydroretinal, 3-hydroxyretinal, and 4-hydroxyretinal were tested (Fig. 2) and all have demonstrated activity in nature. When docked against the squid crystal structure, 11-cis-retinal had the lowest free energy of binding, as expected since this is the functional chromophore for the squid rhodopsin protein. Ranking the energy scores, all-trans-retinal had the highest free energy of binding. For the S. punctatus modeled structure, the lowest energy conformations were observed when bound to 9-cis-retinal isomer, with the next lowest conformations observed with the 11-cis-retinal isomer. The results of the initial pre-Molecular Dynamics (MD) docking screen are provided in Table 1.

Figure 2 Retinaldehyde chromophores used by opsins.

Each of these isomers was used in an in silico docking screen against the S. punctatus homology model and the T. pacificus rhodopsin crystal structure (PDBID 2Z73). (A) 9-cis-retinal (B) 11-cis- retinal (C) 13-cis-retinal (D) all-trans-retinal (E) 3,4-dihydro-retinal (F) 3-hydroxyretinal (G) 4-hydroxyretinal.

Table 1 Autodock results (binding energies in kcal/mol) for the S. punctatus homology model and T. pacificus crystal structure (PDB ID: 2Z73) with retinal isomers before and after molecular mechanics simulations.

Values represent free energy of the lowest scoring conformation (in kcal/mol). “State 0” refers to the model state prior to the start of MD simulations. States “1”, “2”, “3”, and “4” refer to snapshot states immediately after start of MD, and at every 25 ns thereafter. During the course of the simulation, the free energy of binding is minimized for both the 13-cis- and 9-cis-retinal isomers with the S. punctatus homology model. For T. pacificus, State 0 represents the experimentally verified crystal structure, published in complex with 11-cisretinal (Murakami & Kouyama, 2008).

S. punctatus	13-cis -retinal	9-cis -retinal	3,4-dehydro -retinal	3-hydroxy -retinal	4-hydroxy -retinal	all-trans -retinal	11-cis -retinal	
State 0	24.52	6.77	24.06	24.23	24.81	23.44	10.07	
State 1	0.31	−1.56	−2.02	−1.37	−1.30	−1.78	−0.68	
State 2	−1.96	−2.72	−2.15	−2.31	−2.58	−2.41	−1.78	
State 3	−2.11	−0.35	0.43	0.20	0.47	0.42	−2.11	
State 4	−1.76	−1.83	−0.82	−0.48	−1.15	−1.03	−1.49	
T. pacificus	13-cis -retinal	9-cis -retinal	3,4-dehydro -retinal	3-hydroxy -retinal	4-hydroxy -retinal	all-trans -retinal	11-cis -retinal	
State 0	−2.83	−5.43	−4.52	−2.94	−3.20	−4.83	−5.66	
State 1	−0.91	−3.84	−3.66	−3.40	−3.30	−3.78	−1.88	
State 2	−4.57	−2.25	−4.74	−4.24	−4.40	−4.73	−4.17	
State 3	−3.79	−2.43	−2.82	−1.95	−2.56	−2.87	−3.52	
State 4	0.27	−0.80	−1.04	−0.69	−0.75	−1.02	−0.21	

To assess the flexibility of the predicted S. punctatus + 9-cis-retinal complex, molecular dynamics simulations on the opsin and unbound chromophore using AMBER 14 were performed and compared to that of the canonical squid + 11-cis-retinal complex. An overview of the potential energy of two systems during the 100 ns simulation is given in Fig. 3A. While the potential energy of the S. punctatus complex is much lower than that of the squid, both complexes are extremely stable over the long term. Using VMD to plot the RMSD relative to the averaged structure for both complexes also suggests they are stable. For both complexes, these results are given in Fig. 3B. The RMSD of the squid complex begins around 1.5 Å and ends close to 0.7 Å during the simulation, with a mean and standard deviation of 0.87 ± 0.21 Å. The S. punctatus complex fluctuates between 2.56 Å and 8.92 Å, with a mean and standard deviation of 4.67 ± 1.07 Å. Additionally, the per-residue RMSD for both structures remains low during the course of the simulation (Figs. 3B and 3C), with an increase corresponding to the c-terminal loop of the fungal model.

The binding pockets of both receptor proteins were characterized using mdpocket in the fpocket software package (http://fpocket.sourceforge.net) (Le Guilloux, Schmidtke & Tuffery, 2009; Schmidtke et al., 2011). Fpocket generates clusters of spheres to describe pockets identified in a given protein, while mdpocket allows for detection and visualization of pockets on MD trajectories. The pockets predicted for the T. pacificus and S. punctatus MD trajectories are displayed in Figs. 4A and 4B, outlined as both density (red) and frequency (blue) maps. The main binding pockets of both structures can be found at low densities (isovalue of 3) and frequently (isovalue of 50%) during the simulations. In the unbound state, the average of the distances from the center of mass of the retinal ligand to each of the Cα of binding pocket residues in the T. pacificus or S. punctatus structures did not change substantially during the course of the simulation, though there is a slight increase and noticeably more variability in the S. punctatus pocket (Fig. 4E).

Figure 3 Overview plots of MD simulation runs of squid crystal structure with 11-cis-retinal (purple) and S. punctatus model with 9-cis-retinal (gray).

(A) Demonstration that the average RMSD of the fungal structure is higher and more variable than that of the squid, but both are relatively stable during the simulation. (B and C) Average root mean square deviation for each residue position during the 100 ns simulation for S. punctatus model and T. pacificus crystal structure, respectively. Helical regions are annotated with rectangles.

Figure 4 Demonstration of the changes in receptor binding pockets of T. pacificus and S. punctatus structures during MD simulations.

Receptor binding pockets of (A) S. punctatus and (B) T. pacificus structures during MD simulations as predicted by mdpocket (part of the Fpocket package). A density map (red; low isocontour value of 3) is provided, showing the relative enclosure of the binding site. A frequency map (blue; isovalue of 0.5) demonstrates the amount of time (50%) that this pocket was found during the 100 ns MD simulation. The conserved lysine residue is represented in cyan, while the retinal ligand is given in green. (C) Average distance between the retinal ligand center-of-mass and each of the binding pocket residues, as measured in both T. pacificus crystal structure (purple) and S. punctatus homology model (gray) over the course of the 100 ns molecular dynamics simulation.

During the course of the S. punctatus simulation, the 9-cis-retinal ligand shifts approximately 1.6 Å inside the binding pocket of the model. A shift of approximately 1.8 Å by the functional nitrogen atom can be observed during the simulation. The ion lock distance (between E116 and R250) remained consistent, decreasing only slightly from 3.5 Å to 3.4 Å, while the disulfide bond distance (cysteine–cysteine link between C91 and C166) decreased from 5.4 Å to 3.7 Å (Fig. 5). During the T. pacificus simulation, both the 11-cis-retinal ligand shift by less than 1 Å, and the conserved unbound lysine residue (K296) maintains its linear conformation. The T. pacificus ion lock and disulfide distances and orientations remained relatively unchanged, potentially due to the T. pacificus structure being closer to optimal conformation initially.

Figure 5 Change in distances between the features in the model of S. punctatus during 100 ns MD simulation.

(A) cysteine-cysteine disulfide bond and (B) ion lock structural motifs during the S. punctatus 100 ns MD simulation. Initial structural conformations are represented in dark blue with residue designations of “a”. Final conformations are represented in cyan with residue designations of “b”. The distances are plotted over the duration of the simulation in (C) and (D) for cysteine-cysteine and ion lock, respectively.

To assess any potential improvements in docking scores, revised covalent docking was performed using the structures resulting from the previously described simulations and the ligands presented in Fig. 1. Table 1 provides the initial and revised measures of free energy for each docking run, and Table S2 provides energy terms of the ligands and all energy terms for each of the lowest docked runs. For S. punctatus the measures of free energy using the structures from the end of the simulation (frame 3) were lowest when using 13-cis and 9-cis isomers of retinal (−1.76 and −1.83 kcal/mol, respectively), with the 11-cis isomer as the next lowest (−1.49 kcal/mol). For T. pacificus all isomers were relatively similarly low-scoring, although with slight increases using models near the end of the simulation.

Discussion

Using the genomes of early-diverging chytrid fungi B. dendrobatidis and S. punctatus, we identified putative proteins homologous to metazoan Type 2 Rhodopsins. Rhodopsin functions as a photoreceptor via a well-defined interaction between a photon of light, a retinaldehyde chromophore (observed commonly as 11-cis-retinal), and the GPCR opsin protein in order to initiate a cellular response through intracellularly-coupled heterotrimeric G-proteins. There is evidence to suggest that the covalent bond architecture is not biochemically necessary in experimentally manipulated Type 1 opsins (Schweiger, Tittor & Oesterhelt, 1994). However, in naturally occurring opsins this interaction is always facilitated by the presence of a lysine residue in the binding pocket of the GPCR to which the chromophore is covalently bound (Smith, 2010). Of the putative rhodopsin proteins identified in several chytrid fungi, the candidate identified in S. punctatus is the most likely to function as photoreceptor. This protein is highly similar to experimentally verified metazoan rhodopsin proteins and shares structural and functional motifs including most critically the conserved lysine residue within the binding pocket.

Experimental evidence in Blastocladiomycota chytrid fungi indicates they have light regulated behavior (Avelar et al., 2014). Phototaxis has been documented in A. reticulatus and the responsible photoreceptor protein was deduced to be rhodopsin (Saranak & Foster, 1997). Additionally, in the entomopathogenic chytrid fungus Coelomomyces dodgei, photoperiod-dependent spore release has been documented, although the underlying biochemical pathway has not been clearly elucidated (Federici, 1983). The most comprehensive evidence that couples light response behavior and molecular mechanisms is in B. emersonii. Light perception in this fungus requires eye-spot localized photoreceptors that were determined to be fusion proteins of a type 1 rhodopsin and guanylyl cyclase (Avelar et al., 2014; Avelar et al., 2015). There is much less experimental evidence for rhodopsin-regulated behavior in Chytridiomycota. The primary observations are in Rhizophydium littoreum, for which there is evidence of blue-light responsive phototaxis (Muehlstein, Amon & Leffler, 1987), but the underlying molecular mechanisms have not been explored.

In the present study we used in silico docking screens to assess the capacity of the S. punctatus opsin model to bind to known retinal ligands in order to form a functional rhodopsin complex. This sequence is currently the only Type 2 rhodopsin identified in fungi which possesses the conserved lysine and counterion residues, though more complete genomic and transcriptomic sampling of zoosporic lineages will undoubtedly identify additional instances of this gene. Based on this screen, 9-cis-retinal appeared to be the most favorable ligand for use by S. punctatus. As such, the 9-cis isomer was used in subsequent refinement by molecular dynamics. When compared to the squid crystal structure and its canonical 11-cis-retinal ligand, the S. punctatus + 9-cis-retinal complex takes longer to reach a stable conformation, and this conformation deviates quite a bit from the initial structure model. While this suggests inconsistencies with the initial homology model, both the squid and S. punctatus opsin + chromophore complexes appear highly stable. Different chromophores have been observed in nature in opsin complexes being utilized for different purposes. While functional binding pocket residues (e.g., lysine and counterion) are conserved, there are binding pocket residue differences between the squid and fungal structures which could account for the utilization of different chromophores. For example, fewer large hydrophobic residues in the fungal pocket might permit accommodation of different chromophores. Additionally, during the course of exploring why 11-cis-retinal was most often observed in mammalian systems, Sekharan & Morokuma (2011) demonstrated that, generally, 9-cis-retinal is only slightly less stable, and under certain conditions can in fact be more stable, than the 11-cis isomer. Our molecular docking results suggest that one preferential ordering of ligands would be: 9-cis > 13-cis > 11-cis > 4-hydroxy > all-trans > 3,4-dihydro- > 3-hydroxy-retinal. While a thorough treatment of the phylogenetic support for the shared ancestry of these proteins will be presented elsewhere (EM Medina, 2016, unpublished data), the functional relevance of such proteins remains to be explored. The S. punctatus + 9-cis-retinal complex after molecular dynamics simulations supports the hypothesis that this GPCR is a functional photoreceptor and provides a foundation for future work dealing with photoreception in early diverging fungi.

Supplemental Information

Figure S1 PROCHECK-generated Ramachandran plots

A) S. punctatus iTasser homology model and B) T. pacificus rhodopsin x-ray crystal structure (PDB ID: 2Z73).

Click here for additional data file.

Figure S2 3D-1D averaged scores across the length of structures and models

(A) the T. pacificus crystal structure, (B) the S. punctatus homology model against the iTasser GPCR database, and (C) the S. punctatus homology model against the sensory rhodopsin II xray crystal structure from the archaeon Natronomonas pharaonis (PDBid 1H68). Far more residues have averaged scores below 0 in the S. punctatus model using 1H68 as the template than in the other two structures.

Click here for additional data file.

Figure S3 Quality Assessment of of models using ProQM

(A) the T. pacificus crystal structure, (B) the S. punctatus homology model against the iTasser GPCR database, and (C) the S. punctatus homology model against the sensory rhodopsin II xray crystal structure from the archaeon Natronomonas pharaonis (PDBid 1H68). The global quality score for the fungal sequence against the sensory rhodopsin template was lower than that of the iTasser fungal model.

Click here for additional data file.

Table S1 BlastP search results for S. punctatus opsin to PDB

Table of similar protein sequences with structures in the PDB database to the SPPG_00350mod_Long (a curated long version of sequence based on initial genome annotation prediction).

Click here for additional data file.

Table S2 Templates predicted by I-TASSER search of the S. punctatus protein

The columns of this table: Rank, PDB ID, Protein sequence identity, Query protein coverage, Z-score, and sequence chain molecule description

Click here for additional data file.

Table S3 Exploring the binding properties and structural stability of an opsin in the chytrid Spizellomyces punctatus using comparative and molecular modeling energy terms from isomer docking runs

Panel 1 shows the energy and properties of the ligands, panels 2 and 3 shows the lowest energy terms for each retinal isomers from the covalent docking runs on the S. punctatus model or T. pacificus respectively.

Click here for additional data file.

Supplemental Information 1 Exonerate splicing aware protein to genome alignments showing original and corrected gene model

The sequence and report archive provides exonerate alignment results showing the original gene model aligned and post-correction of the genomic sequence. DNA and protein sequences of the corrected model are also provided. The parameters used to run exonerate are included in the report file. This is an archive of data available https://github.com/stajichlab/chytropsin.

Click here for additional data file.

We would like to thank Zhiye Tang and Christopher Roberts for technical assistance. Genome sequence and gene annotations of the Spizellomyces punctatus, Allomyces macrogynus and Batrachochytrium dendrobatidis JEL423 strains were obtained from the Broad Institute and the Origins of Multicellularity Project. Genome of the Batrachochytrium dendrobatidis JAM81 strain was obtained from the Joint Genome Institute Mycocosm database. Computations were performed on the University of California-Riverside Institute for Integrative Genome Biology high performance bioinformatics cluster (http://www.bioinformatics.ucr.edu/).

Additional Information and Declarations

Competing Interests

Author Contributions

Data Availability

The authors declare there are no competing interests.

Steven R. Ahrendt conceived and designed the experiments, performed the experiments, analyzed the data, wrote the paper, prepared figures and/or tables, reviewed drafts of the paper.

Edgar Mauricio Medina conceived and designed the experiments, performed the experiments, analyzed the data, contributed reagents/materials/analysis tools, wrote the paper, reviewed drafts of the paper.

Chia-en A. Chang analyzed the data, contributed reagents/materials/analysis tools, reviewed drafts of the paper.

Jason E. Stajich conceived and designed the experiments, analyzed the data, wrote the paper, reviewed drafts of the paper.

The following information was supplied regarding data availability:

Source code and models are available at https://github.com/stajichlab/chytropsin and also archived at DOI: 10.5281/zenodo.259519.

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
