# Peer review of "Exploring the binding properties and structural stability of an opsin in the chytrid Spizellomyces punctatus using comparative and molecular modeling"

_PeerJ, doi:10.7717/peerj.3206_

## Round 0.1 · original submission · Major Revisions

Please address all of the concerns of both reviewers. I agree that the lack of a bilayer in your simulation strongly hampers your manuscript and that you should include it explicitly (or at least use an implicit model with heterogeneous dielectrics, such as HDGB (Tanizaki and Feig, 10.1021/jp054694f)

Review-level observations by the editor:

The coordinated of the homology model should be provided. Identities of the templates used by I-TASSER (and their level of homology) should alse be provided.

The model does not seem stable at all, as shown by the large increase in RMSD during the simulation. This may be due to the lack of a lipid bilayer in your system, or to the presence of a "wobbly" domain. Could you please add a graph showing average RMSD by aminoacid, to check if that is the case?


Supporting table 1: Units should be included in this table. From the discussion section, I see that those units are kcal.mol. Considering the usual error margin of docking energies, I do not think you can rely on the observed <.5 kcal.mol-1 energy difference between ligands to claim anything regarding the relative binding strengths of the retinal isomers.

Reviewer 1 ·

Basic reporting

This paper describes the modeling of opsin in the spizellomyces punctatus, automated ligand docking, and molecular dynamic to assess the structural and binding properties of an identified opsin-like protein found in Spizellomyces punctatus. The manuscript is generally well-crafted, systematically illustrating the exploration of the binding properties and structural stability of an opsin in the chytrid Spizellomyces punctatus by using comparative molecular modeling. However, the results of the paper is not interesting.

Experimental design

There are some problems with the experiment of Molecular Dynamic because I did not find the membrane setting in the experimental part. For GPCR simulation, the membrane is necessary to get a rational result.

A blast result of opsin in the chytrid Spizellomyces punctatus and animal rhodopsins is necessary.

Validity of the findings

The results are expected but not so attractive.

The results of Molecular Dynamic is incorrect.

Additional comments

The title should be “Exploring the binding properties and structural
stability of an opsin in the chytrid Spizellomyces punctatus using comparative molecular modeling”

Reviewer 2 ·

Basic reporting

In this paper Ahrendt et al describe their attempts at studying a novel potential photosensitive GPCR identified in Fungi. Such a study is very interesting, as we still know very little of GPCRs not expressed in humans. The manuscript is well written, offers a very good and detailed description of the methods used. Unfortunately, the article has several shortcomings that should be corrected before publishing.

1) The authors frequently mention that the Rhodopsin family is currently the family with largest known GPCR members. I think the authors confuse the term with Rhodopsin-like family, or class A family.

2) On page 7 in line 120 the authors write:
"Among the non-Dikarya fungal lineages is a polyphyletic group
" i assume it was probably meant to be written: "
Among the fungal lineages the non-Dikarya is a polyphyletic group"

3) The authors frequently mention the term molecular modeling, when describing techniques that are actually called as "molecular dynamics". Molecular modeling is a term usually coined for techniques used in obtaining 3d protein structures (e.g. homology modeling).

Experimental design

4) The authors mention obtaining a model using the I-Tasser server, it would be good for the paper to also attach the alignment that was used to generate this model. Also seeing as services such as Verify 3d aren't optimized for membrane proteins, it would be good to also score the model using the Proq-M method.

5) Later the authors describe their protocol of carrying out molecular dynamics. Although the results presented are interesting, I strongly believe that using the resources of a computational center, it is possible to achieve more than 10 ns of simulation time. Usually to assess ligand stability and observe modification of the binding site at least 100 ns of simulation time is necessary. The authors also fail to mention whether their simulations where carried out in NVT or NPT conditions. Also the authors don't mention whether the protein was embedded in a membrane, and if yes what was the membrane composition. Seeing as GPCRs are transmembrane receptors, simulations carried out only in water, can't provide accurate results. Also seeing as water has been shown to play an important part in GPCR-ligand binding and activation processes, I believe it would be very beneficial for the authors to repeat the simulations with all-atom water. Input files for such a simulation are easily obtainable using the Charmm-Gui Server.

6) The authors using docking results try to assess the preferred ligand for the newly found protein. It would be beneficial for the paper if they also tried to provide an interpretation why they observe such a difference (e.g. relate it to differences in binding sites).

Validity of the findings

7) Simulation data are not robust. See point (5) above

Additional comments

In this paper Ahrendt et al describe their attempts at studying a novel potential photosensitive GPCR identified in Fungi. Such a study is very interesting, as we still know very little of GPCRs not expressed in humans. The manuscript is well written, offers a very good and detailed description of the methods used. Unfortunately, the article has several shortcomings that should be corrected before publishing.

---

## Round 0.2 · Minor Revisions

I am generally satisfied with the changes performed and with your responses. I believe that a few additional changes to the presentation of your results would considerably increase paper legibility:

Panel A of Fig. 3 is not informative and may be removed without any effect on the strength of your conclusions. Fig. S3, in contrast, should be moved to the main text and discussed there, as it clearly shows that most of the contribution to the high RMSD of the S. puctatus structure (panle B in your Fig. 3) comes from the wobbly C-terminal.

I fear Fig. 4 may be hard to interpret by readers unfamiliar with fpocket output. Depiction of the pockets using mdpocket (http://fpocket.sourceforge.net/manual_fpocket2.pdf) might be more helpful.

Fig. 5 would be strengthened by the inclusion of graphs depicting the evolution of the highlighted distances throughout the simulations.

Reviewer 1 ·

Basic reporting

This revision also met most questions

Experimental design

The MD simulations seems acceptable.

Validity of the findings

no comment.

Additional comments

some typos should be corrected:
Such as: line 395: GCPR should be GPCR

---

## Round 0.3 · accepted · Accept

You have most satisfactorily addressed all remaining points, and I am glad to approve you paper for publication.